# Analysis of the Effects of the *Vrn-1* and *Ppd-1* Alleles on Adaptive and Agronomic Traits in Common Wheat (*Triticum aestivum* L.)

**DOI:** 10.3390/plants13111453

**Published:** 2024-05-23

**Authors:** Kirill O. Plotnikov, Alexandra I. Klimenko, Ekaterina S. Ovchinnikova, Sergey A. Lashin, Nikolay P. Goncharov

**Affiliations:** 1Early Maturity Genetics Laboratory, Institute of Cytology and Genetics, Siberian Branch of the Russian Academy of Sciences, Akademika Lavrentieva Avenue, 10, 630090 Novosibirsk, Russia; 2Kurchatov Genomics Center, Institute of Cytology and Genetics, Siberian Branch of the Russian Academy of Science, Akademika Lavrentieva Avenue, 10, 630090 Novosibirsk, Russialashin@bionet.nsc.ru (S.A.L.)

**Keywords:** common wheat, *Triticum aestivum* L., *Vrn-1* alleles, *Ppd-1* alleles, earliness, yield, near-isogenic lines, principal component analysis

## Abstract

Wheat heading time is primarily governed by two loci: VRN-1 (response to vernalization) and PPD-1 (response to photoperiod). Five sets of near-isogenic lines (NILs) were studied with the aim of investigating the effect of the aforementioned genes on wheat vegetative period duration and 14 yield-related traits. Every NIL was sown in the hydroponic greenhouse of the Institute of Cytology and Genetics, SB RAS. To assess their allelic composition at the VRN-1 and PPD-1 loci, molecular markers were used. It was shown that HT in plants with the *Vrn-A1vrn-B1vrn-D1* genotype was reduced by 29 and 21 days (*p* < 0.001) in comparison to HT in plants with the *vrn-A1Vrn-B1vrn-D1* and the *vrn-A1vrn-B1Vrn-D1* genotypes, respectively. In our study, we noticed a decrease in spike length as well as spikelet number per spike parameter for some NIL carriers of the Vrn-A1a allele in comparison to carriers of the *Vrn-B1* allele. PCA revealed three first principal components (PC), together explaining more than 70% of the data variance. Among the studied genetic traits, the *Vrn-A1a* and *Ppd-D1a* alleles showed significant correlations with PCs. Regarding genetic components, significant correlations were calculated between PC3 and *Ppd-B1a* (−0.26, *p* < 0.05) and *Vrn-B1* (0.57, *p* < 0.05) alleles. Thus, the presence of the *Vrn-A1a* allele affects heading time, while *Ppd-D1a* is associated with plant height reduction.

## 1. Introduction

Common wheat (*Triticum aestivum* L., genome BBAADD, 2*n* = 6*x* = 42) is one of the most important food crops, feeding up to 35% of the world’s population [1,2,3]. As the world’s population is expanding, demand for wheat is only going to rise [4,5]. However, environmental constraints associated with climate change, such as increased temperatures, drought stress, or excessive rainfall, followed by fungal outbreaks and, in general, an increased frequency of unexpected extreme weather events, are considered serious threats to the world’s wheat production [6,7,8,9].

Wheat adaptability to different environments and yield potential depend crucially on its phenology and ability to regulate heading time, the transition from vegetative to generative development [10,11,12]. Thus, wheat’s late heading is advantageous for regions with longer growing seasons, like central or northern parts of Europe, the United Kingdom, the Middle Volga Region of the Russian Federation, etc. On the other hand, for regions with short growing seasons, like the north of Canada or Western Siberia, early heading helps to avoid late spring and early autumn frost damage and therefore secures a high yield [10,13,14,15]. With global warming acceleration, wheat heading time adjustment towards earliness might become an important breeding strategy also for agro-climatic zones characterized by longer growing seasons, presuming it will facilitate the avoidance of mid-summer heat waves, infection manifestations, excessive rainfall, or freezing events in early autumn [9,16,17,18,19].

In wheat, the duration of the vegetative period is controlled to a large extent by vernalization (*Vrn-1*) and photoperiod response (*Ppd-1*) genes and earliness *per se* (*Eps*) genetic factors [20,21,22]. The latter works on fine-tuning wheat heading time once vernalization and photoperiod requirements are met [20,23]. *Eps* can be characterized as intrinsic earliness and is believed to be independent from environmental clues such as vernalization or day length but responsive to temperature changes [23,24]. This system has not been extensively explored, so far mainly in the cultivated diploid wheat *Triticum monococcum* L. [24,25,26].

Common wheat vernalization response and spring growth habit are primarily under the control of alleles at the VRN-A1, VRN-B1, and VRN-D1 loci, localized on the long arms of homologous chromosomes 5A, 5B, and 5D, respectively [27,28]. Allelic combinations of the *Vrn-1* gene determine whether vernalization requirements (the prolonged exposure to low temperatures) in wheat have to be fulfilled. The presence of at least one dominant *Vrn-1* allele (*Vrn-A1*, *Vrn-B1*, *Vrn-D1*) elicits spring growth habits regardless of the allelic state of other genes, while the combination of all recessive *Vrn-1* alleles (named *vrn-A1*, *vrn-B1*, *vrn-D1*) results in vernalization-requiring or winter wheats [29,30,31,32,33]. Genetic variations of VRN-1 loci are also recognized as important regulators of wheat heading time, sequenced as follows by their potency: *Vrn-A1* > *Vrn-D1* > *Vrn-B1*. Thus, cultivars with the dominant *Vrn-A1* allele flower earlier than those with the dominant *Vrn-B1* or *Vrn-D1* alleles, and plants with the dominant *Vrn-B1* allele show the latest flowering time of them all [34,35,36,37]. Due to their effect on growth and development, *Vrn-1* genes play an important role in wheat adaptation and yield formation in different environments [10,12,37]. Therefore, a variety of combinations of *Vrn-1* alleles securing wheat adaptability and maturity were determined for different agro-climatic zones [38,39,40,41]. Thus, according to a survey concerning European spring wheat accessions, the highest average yield per plant was recorded for *Vrn-A1* and *Vrn-A1+Vrn-B1* genotypes, while early-ripening carriers of all three *Vrn-1* dominant alleles suffered a yield penalty [34]. The dominant *Vrn-D1* allele is mostly present in tropical and sub-tropical regions [42,43]. The majority of Turkish cultivars studied by Andeden et al. [44] possessed one or more dominant *Vrn-B1* and *Vrn-D1* alleles. In 278 Chinese common wheat accessions studied by Zhang et al. [41] with respect to *Vrn-1* allelism, the *Vrn-D1* spring growth allele was prevalent, and all early heading cultivars were carriers of the strongest *Vrn-A1* allele either alone or in combination with any other dominant *Vrn-1* allele.

Once vernalization requirements are met, daylight duration is another important factor in wheat flowering induction. Thus, photoperiod insensitive or day neutral wheat varieties flower rapidly in long (~16 h day length) and short days (<12 h day length), while wild-type photoperiod-sensitive cultivars flower rapidly in long days and later in short days [16,45,46]. This sensitivity of wheat to photoperiod is mainly regulated by *Ppd-1*, a pseudo-response regulator (PRR), mapped on 2A (*Ppd-A1*), 2B (*Ppd-B1*), and 2D (*Ppd-D1*) chromosomes [47,48]. Dominant alleles (photoperiod insensitive) are assigned by an “a” suffix (e.g., *Ppd-A1a*, *Ppd-B1a*, *Ppd-D1a*), while recessive alleles, photoperiod sensitive, are assigned by an “b” suffix (e.g., *Ppd-A1b*, *Ppd-B1b*, *Ppd-D1b*) [45,49]. So far, several insensitive *Ppd-1* alleles have been described in common wheat [50,51,52]. It was reported that the *Ppd-D1a* allele has the strongest effect on common wheat photoperiod sensitivity and early heading, followed by *Ppd-B1a* and *Ppd-A1a* alleles, ranked in the following order: *Ppd-D1a* > *Ppd-B1a* > *Ppd-A1a* [16]. *Ppd-1a* alleles are widespread around the world, helping to improve wheat cultivars adaptability in a range of environments. Thus, in southern Europe, Argentina, and North Africa, accelerated heading caused by photoperiod insensitivity is highly beneficial for yield stability as it allows plants to escape high temperatures during grain-filling stages [16,40,53,54]. In Japan, earliness conferred by *Ppd-1a* alleles helps to avoid the spread of fungal diseases and pre-harvest sprouting caused by heavy rainfalls [50,55]. *Ppd-1a* allele frequency also decreases gradually from countries at low latitude to countries at high latitude where late season stress normally does not occur (such as central parts of Europe, the United Kingdom, and the central regions of the Russian Federation) [17,56,57]. However, it was already recognized that, due to increased seasonal weather fluctuations and adverse conditions caused by climate change, the introduction of *Ppd-1a* alleles and the development of early-flowering winter wheat cultivars in the United Kingdom might become an important part of the food safety mitigation strategy [19].

In order to address environmental challenges and fulfill the world’s growing demand for wheat, further investigations into the molecular genetics of plant heading time regulation and its association with yield-related traits have to be performed. Near isogenic lines (NILs) are an important tool used in crop genetics for detailed mapping and functional analysis of one or a few specific loci. Within one set, all NILs have a genetically identical background except for specific targeted genetic locus/loci. This means that virtually any phenotypic differences recorded between NILs will be due to targeted locus/loci, and therefore phenotype-genotype associations can be established [58]. Thus, one of the most well-known sets of NILs developed by Dr. Pugsley in 1971 in the genetic background of Australian spring wheat cv. Triple Dirk (TD) was extensively used in a variety of studies of wheat genetics and phenology [32,33,59,60,61]. Moreover, the availability of several sets of NILs for certain locus/loci would give an additional advantage, as it would also mean an opportunity to test and validate the phenotypic impact attributed to this locus/loci under different genetic backgrounds. Despite being an important breeding and genetic tool, there are certain concerns related to NIL research. First of all, the development of NILs is a highly labor-intensive and time-consuming process. Secondly, donor’s genetic material contamination is one of the most important methodological problems in NILs development [58,62]. It was discovered by Zeven et al. [62] that genetic resemblance between existing NILs and their recurrent parental cultivars might not always occur. Moreover, NILs might exist in more than one genotype, and this issue required researchers’ attention and additional genetic testing [62]. It is noteworthy that for the majority of NILs used in this study, the exact allelic combination of *Vrn-1* and *Ppd-1* genes was never determined with diagnostic molecular markers, and genotype was only assumed based on observed phenology.

The use of multiple NILs and their thorough phenotyping also often leads to the generation of massive data sets with a large number of variables, which is difficult to comprehend. Principal component analysis (PCA) is a statistical tool frequently used to address this issue [63,64]. The PCA approach helps to reduce dataset complexity and to transform multiple genetic, phenotypical, and agronomic traits into a few principal components (PCs), followed by finding the most important correlations between them and dissecting the noise produced by the large volume of data [15,63,64,65,66].

Although the dynamics and regulation of wheat development are still not very well defined processes controlled by combinations of genetic and environmental factors, it is presumed that the length of developmental stages and their transition from one to another affect grain yield-related traits [67,68]. Thus, understanding the role of wheat developmental genes *Vrn-1* and *Ppd-1* in wheat phenology and yield formation can help breeding programs improve climate change adaptation. However, under field experimental conditions, a variety of environmental factors experienced at different locations in different years often cause considerable results variability [12].

Therefore, the aim of this study was to evaluate the effect of different combinations of *Vrn-1* and *Ppd-1* alleles on heading time and major agronomic traits in common wheat in a controlled and stress-free greenhouse environment. To address this objective, we performed phenotypic and genotypic characterization of five different sets of NILs, followed by PCA and correlation analysis, to find patterns influencing wheat phenology and yield-forming traits.

## 2. Results

### 2.1. Verification of Spring Genotypes of Studied NILs

In this study, we used five sets of NILs on the VRN-1 loci, 14 NILs in total (Table 1). To the best of our knowledge, no pedigree overlap was recorded between cultivars used as recurrent parents during NIL production (Figure 1a) [69]. All studied NILs were grown in a stress-free greenhouse environment under long days without prior vernalization. In order to demonstrate major phenotypical features of the studied plants, TDD, Sk3b (*Vrn-A1*), JF (*Vrn-A1*), M808 (*Vrn-A1*), and Pr (*Vrn-A1*) NILs were pictured at the 38th day after seeding (Figure 1b).

To overcome uncertainty related to NIL development, allelic combinations of *Vrn-A1*, *Vrn-B1*, and *Vrn-D1* genes were verified for all studied sets of NILs with the use of previously published diagnostic molecular markers (Appendix A). Briefly, DNA markers developed by Yan et al. [30] were used for identification of the dominant *Vrn-A1a* (PCR fragments 965 and 876 bp) and *Vrn-A1b* (714 bp fragment) alleles characterized by the presence of insertions or deletions in the *Vrn-A1* promotor region, as well as for identification of the recessive *vrn-A1* allele (734 bp fragment). Integrity of the *Vrn-A1* intron 1 region was checked with primers Intr1/C/F and Intr1/AB/R (1068 bp fragment), developed by Fu et al. [32]. DNA markers based on the presence or absence of large deletions within intron 1 of the *Vrn-B1* and *Vrn-D1* genes were developed by Fu et al. [32] and allowed us to specify the *Vrn-B1*, *vrn-B1*, *Vrn-D1,* and *vrn-D* alleles in the NILs2.2 studied for the determination of the structure of the *Vrn* gene.

For all studied NILs, allelic composition at VRN-1 loci corresponded to the presumed ones (Figure 1c and Figure 2a,b, Table 1). Moreover, these *Vrn-1* alleles coincided with those identified in TDD, TDB, and TDE, NILs that originally served as donors of *Vrn-1* dominant alleles for studied NILs, ensuring a lack of contamination during NIL generation and further reproduction (Figure 1c and Figure 2a,b).

### 2.2. Identification of the Ppd-1 Allelic Composition of Studied NILs

Since photoperiod response is one of the major controllers of wheat early heading, we also performed a systematic check of *Ppd-1* allelic composition in the studied NILs.

As was shown before, large deletions upstream of the *Ppd-A1* and *Ppd-D1* coding regions are associated with photoperiod insensitivity. Molecular markers developed by Nishida et al. [50] and Beales et al. [51] were used to detect these mutations and to discriminate carriers of photoperiod insensitive alleles among the studied NILs. The integrity of the *Ppd-A1* promoter region was checked, and a 299 bp fragment was detected in all studied NILs, meaning the presence of the photoperiod-sensitive allele *Ppd-A1b* (Figure 3(a1)). With respect to the B genome, the allele variant related to the insertion in the *Ppd-B1* promoter region was checked with PCR primers developed by Nishida et al. [50], and for all studied cultivars, fragments characteristic of the recessive *Ppd-B1b* allele were detected (Figure 3(a2)). In accordance with previously published studies, to determine *Ppd-B1* copy number variations (*Ppd-B1* gene copies arranged in tandem) three sets of primers were used [52,57]. Primers S64-copy-F1/S64-copy-R1 were used to identify the “Sonora 64/Timstein”-type *Ppd-B1a* photoperiod insensitive allele based on intercopy junction detection (223 bp PCR fragment). A DNA sample of cv. Sonora 64 was used as a positive control. Primers CS-copy-F1/CS-copy-R1 and PpdB1_F25/PpdB1_R70 were used to identify the Chinese Spring-type *Ppd-B1c* allele by the presence of a truncated copy (425 bp fragment) and an intercopy junction (994 bp fragment). A DNA sample of cv. Chinese Spring was used as a positive control. Based on PCR results, fragment-specific for *Ppd-B1a* allele was detected in the following NILs: TDD, TDB, and TDE, as well as in Pr (*Vrn-D1*) (Figure 3(a2)). No truncated copy of *Ppd-B1* gene as well as no *Ppd-B1c* allele was identified in any of the studied cultivars (Figure 3(a2)). PCR markers developed by Beales et al. [51] were used to identify a 2089 bp deletion in the promotor region conferring the photoperiod insensitivity of the *Ppd-D1* gene. Results of this test identified that all NILs (Sk3b) carried the dominant day-length insensitive allele *Ppd-D1a* (288 bp fragment). Meanwhile, JF and M808 NILs are all carriers of the day-length sensitive allele *Ppd-D1b* (414 bp fragment). For Priboi NILs, differences in *Ppd-D1* alleles between lines were detected. Thus, Pr (*Vrn-A1*) and Pr (*Vrn-B1*) are carriers of the *Ppd-D1a* insensitive allele, while Pr (*Vrn-D1*) carries *Ppd-D1b* (Figure 3(a3)).

A summary of *Vrn-1* and *Ppd-1* allelic combinations for all five sets of NILs can be found in Table 1 and Table 2.

### 2.3. Influence of Ppd-1 and Vrn-1 Allelic Combinations on Wheat Earliness

Differences in heading time (HT) between NILs were studied in several steps. First of all, an HT comparison was performed within each set of NILs. For each set of NILs, carriers of the *Vrn-A1a* allele demonstrated significantly reduced HT in comparison to NILs that possessed *Vrn-B1* or *Vrn-D1* dominant alleles.

As a next step, we compared the times of heading for all NIL carriers of the *Vrn-A1a* allele. As shown in Figure 4a, there was no difference in HT among these NILs, except for the 2.9 day (*p* < 0.05) difference between Sk3b (*Vrn-A1*) and JF (*Vrn-A1*).

Interestingly, once the same analysis was performed for all NIL carriers of *Vrn-B1* alleles (Figure 4b), significant differences in HT were found between Sk3b (*Vrn-B1*) and M808 (*Vrn-B1*), 15 days *p* < 0.001. Although both lines Sk3b (*Vrn-B1*) and Pr (*Vrn-B1*) are carriers of the *Ppd-D1a* allele, a significant difference in HT (10.8 days, *p* < 0.001) was recorded between them as well. Within the same set, HT differences of 10.7 days were found between TDB and M808 (*Vrn-B1*), and 6.6 days were found between TDB and Pr (*Vrn-B1*). The difference in HT between Pr (*Vrn-B1*) and M808 (*Vrn-B1*) was 4.1 days (*p* < 0.05).

The same trend was observed for NILs with the *vrn-A1vrn-B1Vrn-D1* genotype (Figure 4c). Significant differences in HT (7.2 days, *p* < 0.05) were recorded between carriers of dominant *Ppd-1a* alleles (Sk3b (*Vrn-D1*) and TDE) and carriers of *Ppd-1b* recessive alleles (M808, *Vrn-D1*) and (JF, *Vrn-D1*), as well as between JF (*Vrn-D1*) and M808 (*Vrn-D1*). It is noteworthy that with any *Vrn-1* genetic background, no significant differences were found between the HT of NILs (Sk3b) and NILs (TD), carriers of the *Ppd-D1a* and *Ppd-B1a* alleles.

Moreover, by combining NILs into three groups in accordance with the dominant *Vrn-1* allele, it was shown that HT in plants with the *Vrn-A1vrn-B1vrn-D1* genotype was reduced by 29 and 21 days (*p* < 0.001) in comparison to HT in plants with the *vrn-A1Vrn-B1vrn-D1* and *vrn-A1vrn-B1Vrn-D1* genotypes, respectively (Figure 4d).

### 2.4. PCA and Pearson’s Correlation Analysis to Study the Effect of Ppd-1 and Vrn-1 Allelic Combinations on Phenology and Agronomic Traits of NILs

The PCA approach was used to simplify data interpretations and to reduce the high number of variables into a few principal components. In our study, out of a total of 14 PCs, the first 10 explained more than 98% of the variation (Table 3). Following the Kaiser rule, only PCs with eigenvalues higher than 1 were kept for further analysis. The first three PCs had eigenvalues > 1 (Table 3). These three PCs explained more than 70% of the total variability, where PC1 accounts for 41.58% of the variance, while PC2 and PC3 account for 16.6% and 13.84%, respectively (Table 3).

According to Table 3, PC1 gives the highest and roughly equal weight to the PH, SL, SNS, GNS, and GWS variables and can be interpreted as a measure of plant architecture and yield potential. Pearson’s correlations between the original variables and each of the principal components were computed to further deepen PC interpretation (Appendix A, Figure 5). This analysis shows a strong correlation between PC1 and PH, SL, and yield-related traits. On the other hand, PC1 negatively correlates with the spring allele *Vrn-A1a* (−0.4, *p* < 0.05) and the photoperiod insensitive allele *Ppd-D1a* (−0.58, *p* < 0.05), presuming that the presence of these alleles in wheat germplasm will correlate with changes in morphology (PH, SL) and a reduction in yield potential in the stress-free environment studied. The variation captured by PC2 was mainly due to SF (calculated as the number of grains per spikelet), GNS, GWS, and HT, while the correlation with SF was the highest (0.75, *p* < 0.05), followed by GNS (0.53, *p* < 0.05), GWS (0.47, *p* < 0.05), and HT (−0.5, *p* < 0.05), according to Table 3 and Appendix A, Figure 5. Therefore, PC2 can be viewed as a measure of the SF of early heading plants. Additionally, PC2 is positively correlated with the presence of the spring allele *Vrn-A1a* (0.37, *p* < 0.05) and the photoperiod insensitive allele *Ppd-D1a* (0.46, *p* < 0.05) but negatively correlated with *Vrn-D1* (−0.28, *p* < 0.05). A significant negative correlation between PC2 and *Ppd-B1a* (−0.37) was also recorded (Appendix A, Figure 5). Correlations between PC2 and PH, SNS, and SL did not pass the Bonferroni correction to be significant (Figure 5). Variability is explained by the third principal component, which is comprised principally of HT and awnlessness, and no significant correlations were recorded between PC3 and yield-related traits (Table 3 and Appendix A, Figure 5). Regarding genetic components, significant correlations were calculated between PC3 and *Ppd-B1a* (−0.26, *p* < 0.05) and *Vrn-B1* (0.57, *p* < 0.05) alleles. Interestingly, unlike other principal components, PC3 has opposing correlations with *Vrn-A1a* (−0.58, *p* < 0.05) and *Ppd-D1a* (0.6, *p* < 0.05) alleles (Appendix A, Figure 5). Therefore, unlike PC2, PC3 is a measure of plant earliness occurring due to the interaction of *Vrn-A1a* and photoperiod alleles different from *Ppd-D1a*.

Plants from different NIL sets were scattered in the four quarters of the PCA plots (Appendix A), indicating a high level of variability between tested NILs. However, on the PC1 vs. PC2 plot, plants belonging to NILs (Sk3b) and Pr (*Vrn-A1*) were mostly located in the quarter with the highest PC2 and the lowest PC1, while plants from JF and M808 NILs, specifically JF (*Vrn-D1*), M808 (*Vrn-B1*), M808 (*Vrn-D1*), and TDB, were located in the area with the highest PC1 and the lowest PC2 (Appendix A). The rest of the NILs tend to be located around the origin of the graph. Based on the PC1 vs. PC3 and the PC2 vs. PC3 plots, we can see that plants TDD, M808 (*Vrn-A1*), and JF (*Vrn-A1*) are located in areas with the lowest PC3, while Pr (*Vrn-B1*) and Sk3b (*Vrn-B1*) are located in areas with the highest (Appendix A). In the analyzed bi-blots, NILs with the highest PC2 are characterized by the presence of *Ppd-D1a*, an allele securing photoperiod insensitivity, while NILs with the highest PC1 are all carriers of recessive alleles *Ppd-1b* and *vrn-A1* and were clustered towards increased HT, SL, SNS, and a high number of tillers.

While the *Vrn-A1a* allele showed a significant correlation with HT (−0.8, *p* < 0.05), the presence of the *Ppd-D1a* allele negatively correlated with PH (−0.65, *p* < 0.05) and SL (−0.6, *p* < 0.05). Opposed to *Ppd-D1a*, a significant positive correlation was found between the *Vrn-D1* allele and PH (0.31, *p* < 0.05). There was no interaction found between PH and the *Vrn-B1* allele; however, the presence of *Vrn-B1* alleles might affect PH indirectly via prolongation of the vegetative period due to their positive interaction with HT (0.46, *p* < 0.05). HT and PH both had strong positive correlations with SL and SNS. According to the PCA results, there is also a strong positive correlation between acceleration of heading time and agronomic traits, especially SF. Therefore, shorter, early heading NILs, carriers of *Ppd-D1a* alleles, might potentially compensate for yield loss in comparison to late-developing wheats via an SF improvement. Negative correlations were recorded between *Vrn-A1a* and GWS, as well as between *Ppd-D1a* and GWS (Appendix A). Although these correlations failed to reach statistical significance under Bonferroni correction for the whole data set, according to the ANOVA test, it was noticed that for NILs Sk3b (*Vrn-A1*) and Pr (*Vrn*-*A1*), the measured GWS was significantly lower than for NILs Sk3b (*Vrn-B1*) and Pr (*Vrn-B1*), respectively. Strong positive correlations between GWS and GNS traits were noticed, meaning that under stress-free conditions, an increase in GNS is accompanied by higher GWS, and no trade-off between grain number and weight was recorded. A negative correlation between *Ppd-D1a* and tillering was also recorded but did not pass the Bonferroni correction to be significant. Unlike for the *Ppd-D1a* allele, for *Ppd-B1a*, no strong direct correlations with phenotypical or yield-related traits were discovered. However, the presence of *Ppd-B1a* contributed to the data variations reflected by PC2 and PC3.

## 3. Discussion

Ensuring the yield stability of cereals under changing environmental conditions is a way to mitigate malnutrition and food security issues rising worldwide. Wheat adaptation to different agro-climatic zones is to a great extent correlated with the duration of the vegetative period, which is majorly governed by two genes: *Vrn-1* and *Ppd-1* [70]. In this research, five sets of NILs were studied in a controlled and stress-free greenhouse environment. Noteworthy, although donors of spring *Vrn-1* alleles were the same, studied NILs were developed on the basis of cultivars with no relationships across pedigrees, providing us with an opportunity to check the impact of a specific allele within a variety of genetic backgrounds. In this study, for the first time, in addition to performing *Vrn-1* and *Ppd-1* genotyping, computational approaches were used to explore the effect of various allelic combinations of these genes on the adaptive and agronomic traits of utilized NILs. In the stress-free environment studied, PH and SL showed strong positive correlations with each other and with yield-related traits. At the same time, strong, significant relationships were found between genetic components and plant phenology. However, none of the studied genetic components had strong correlations with yield-forming traits such as GNS, GWS, and SF. Although significant negative correlations were recorded between SNS and *Vrn-A1a* as well as *Ppd-D1a*, both dominant alleles were involved in wheat heading time acceleration under the studied conditions. These results are in line with previously reported data [16,34,35,71]. Our results suggest that *Vrn-A1a* and *Ppd-D1a* are important genetic factors driving changes in plant architecture, yield performance, and phenology. Interestingly, the presence of *Ppd-1a* alleles affects plant heading times and plant height, even under long day setting. It was also noticed that penalties on yield-related components, which had been so often recorded for early heading cultivars, also occurred under the studied conditions, although the results of previously conducted field experiments suggested a decrease in average grain weight as grain number increased [72]. The number of tillers per plant is another important agronomic trait. In our study, a significant negative correlation was recorded between the *Vrn-A1a* allele and plant tillering capacity, which is in line with the report of [73], where the presence of the *vrn-A1* allele was associated with a higher number of tillers.

To begin with, *Vrn-1* and *Ppd-1* gene allelic compositions were determined with allele-specific DNA markers. For each NIL, the *Vrn-1* allelic composition coincided with the presumed one and with the one found in appropriate NILs (TD). Additionally, *Ppd-1* allele diversity was checked. As a result, studied NILs were divided into three groups based on their photoperiod sensitivity: strongly sensitive NILs (JF) and NILs (M808), carriers of all three photoperiod sensitive alleles (*Ppd-A1b Ppd-B1b Ppd-D1b*), moderately insensitive NILs (TD), carriers of the *Ppd-B1a* photoperiod insensitive allele (*Ppd-A1b Ppd-B1a Ppd-D1b*), and strongly insensitive to photoperiod NILs (Sk3b) and NILs (Pr) possessing the *Ppd-D1a* allele (*Ppd-A1b Ppd-B1b Ppd-D1a*). These results are in line with previously reported data by Fait and Balashova [74], who identified *Ppd-D1a*-based photoperiod insensitivity of Skorospelka 3b and Priboi winter wheat cultivars as well as day length sensitivity of Mironovskaya 808. For the first time, to our knowledge, the *Ppd-1* genotype of NILs (TD) was checked with diagnostic molecular markers. We found that TDD, TDE, and TDB NILs are all carriers of the photoperiod-insensitive “Sonora64/Timstein”-like *Ppd-B1a* allele. This finding is particularly important, firstly because it helps to explain the molecular basis of the photoperiod insensitivity of NILs (TD), and secondly, due to the accessibility and popularity of these NILs among breeders and researchers, further investigations could be easily performed. Additionally, we also found that unlike NILs Pr (*Vrn-A1*) and Pr (*Vrn-B1*), NIL Pr (*Vrn-D1*) carries the *Ppd-B1a* allele, the same as was found in TDE. This incongruence may be related to the donor’s genetic material contamination, a phenomenon often occurring during NILs development [58].

Subsequent to the *Vrn-1* and *Ppd-1* allelic combination evaluation, the phenology and agronomic traits of NILs were investigated. In our experimental setup, the *Vrn-A1a* allele acted as the main accelerator of wheat earliness, irrespective of NIL genetic background. NILs with *Vrn-A1a vrn-B1 vrn-D1* genotype showed significantly earlier heading in comparison to carriers of *Vrn-B1* or *Vrn-D1* alleles, which is in line with reports of [34,35,36,37] and previously published data of [58,75] who used the same setup of NILs. However, among Chinese spring wheat varieties studied by Zhang et al. [41], the early heading phenotype was also associated with the presence of the *Vrn-A1* allele, while the latest heading was associated with *Vrn-D1*, and cultivars that possessed *Vrn-B1* showed intermediate values. Regarding the *Vrn-1* genotype in relation to yield performance, a decrease in spike productivity parameters such as spike length, spikelet number, and grain number was shown in the same setup of NILs in the following order: *Vrn-B1 > Vrn-D1 > Vrn-A1* [58,75]. In our study, we also noticed a decrease in spike length as well as spikelet number per spike parameter for some NIL carriers of the *Vrn-A1a* allele in comparison to carriers of the *Vrn-B1* allele; however, differences between *Vrn-A1a* and *Vrn-D1* were not pronounced. Interestingly, according to [58], small plot field experiments performed with NILs (M808) in Western Siberia showed an equal yield for M808 (*Vrn-A1*) and M808 (*Vrn-D1*) NILs and a considerable decrease in yield performance for M808 (*Vrn-B1*), pointing out the importance of plant genotype adaptability in certain environments. Additionally, in our study, plants with the *Ppd-1a* genotype generally showed accelerated heading in comparison to carriers of the *Ppd-1b* genotype, meaning that even under long photoperiods characteristic of higher and middle latitudes, *Ppd-1a* alleles confer wheat earliness. The differences in heading time were also recorded between different NIL (*Vrn-B1*) carriers of the *Ppd-1b* genotype. Recorded variations are NIL genetic background-dependent and most likely attributed to the *Eps* system acting specifically in each line and probably affecting *Ppd-1* and *Vrn-1* effectiveness. These differences should be further explored in more detail to advance our knowledge about the specific interactions of these three major genetic systems.

To ensure an adequate interpretation of the large dataset obtained, principal component analysis (PCA) and correlation analysis were performed in our study. PCA is an important tool frequently used in plant genetics to compare a large number of different cultivars based on their genetics and agronomic traits, countries of origin, etc. in order to find a specific pattern via data dimensionality reduction [15,65,66]. The PCA approach revealed three main principal components (PC) together explaining more than 70% of the data variation, where PC1 can be interpreted as a measure of plant habitus and habitus-related agronomic traits, PC2 as an earliness and spike fertility, and PC3 as a measure of wheat heading time and awnlessness. In our study, awnlessness also positively correlates with GNS and GWS, which is in line with a number of previous observations showing that in favorable environments, awnless cultivars are either superior or equal to the awned ones with respect to yield or yield-related traits, while awned cultivars are advantageous for heat-stressed regions [76]. However, according to the recent analysis performed by Sanchez-Bragado et al. [77], there is no clear effect of awn presence on wheat yield across a variety of environments.

Correlation analysis was carried out subsequently for PCA and allowed us to check not only correlations occurring between agronomic traits but also between traits and identified PCs, looking at each PC as an additional artificially generated “trait”. This is particularly important since, unlike PC1 and PC2, none of the studied genetic traits had strong correlations with yield-related traits such as GNS, GWS, or SF, proving once again the genetic complexity of yield as a trait [78,79]. However, *Vrn-A1a* and *Ppd-D1a* alleles showed significant correlation with all three PCs, assuming a strong connection of these alleles and plant phenology and architecture, while *Vrn-B1* only correlated with PC3, and *Vrn-D1* showed moderate correlation with PC1 and PC2, both components highly correlating with yield-related traits under studied conditions. The latter one is in line with the previously reported results of [35,38,80], which suggested that the presence of the *Vrn-D1* allele is directly or indirectly related to a higher yield and the proposed introduction of the *Vrn-D1* allele into elite breeding material. The correlation of *Vrn-B1* solely with PC3 might indicate a specific role of this allele in the regulation of vegetative period duration under near-ideal greenhouse conditions. There is also a strong negative correlation between plant height and the presence of *Ppd-D1a*, which was also recorded, supporting the previous findings of [81,82]. Additionally, research carried out by Steinfort et al. [68] on a setup of NILs showed that photoperiod sensitivity at *Ppd-D1* lengthened the stem elongation phase while delaying flowering. In turn, stem elongation duration affects spike growth period, number of fertile florets, and hence the number of grains per spike and overall yield via the duration of nutrient supply assimilation [68,83,84]. Recorded in this study are negative correlations between *Ppd-D1a* and SL. SNS are also in line with the abovementioned observation and previously published data regarding the *Ppd-D1* effect on spike architecture and yield formation [16,85]. It is worth pinpointing that, unlike *Ppd-D1a*, the presence of the *Ppd-B1a* allele showed a moderate negative correlation only with GNS and PC2, PC3, presuming a modest negative effect of this allele on plant yield-related traits and phenology under studied conditions. Additionally, with respect to yield, according to field experiments performed by Arjona et al. [86], the presence of the *Ppd-B1a* allele did not impact the grain yield of durum wheat (*T. durum Desf.*) cultivars. In our research, the presence of *Ppd-B1a* also did not correlate with changes in plant height or spike architecture. During field experiments described by Worland et al. [16], it was shown that accessions possessing the dominant *Ppd-B1* allele were shorter in comparison to photoperiod-sensitive ones, although the effect of *Ppd-D1a* was still more pronounced.

Thus, the presence of the *Vrn-D1* allele offers a yield advantage in regions with longer growing seasons, which is in line with the typically high incidence of the *Vrn-D1* allele in Chinese, Japanese, and Pakistani wheat varieties, as well as in wheat accessions from Southern Europe [40,41,87,88]. On the other hand, one of the most successful spring wheat Canadian cultivars (i.e., cv. Marquis) possessed the *Vrn-A1a* allele alone or in combination with *Vrn-B1*, ensuring these cultivars early heading during short growing seasons in the Canadian high-latitude northern regions [13,14,89]. The same is true for cultivars specific to the Western Siberia region, as shown by Efremova et al. [90]. The presence of the *Ppd-D1a* allele conferred substantial improvement on grain yield and related traits in environments where drought and severe heat waves are common. Thus, according to [16], *Ppd-D1a* allele introduction into common wheat germplasm led to over 35% of the yield advantage in southern European regions, while in Germany and England, the annual yield varied from +9% to −16%. Research carried out by Foulkes et al. [91] showed that the presence of the *Ppd-D1a* allele in winter cultivars appeared to be neutral with respect to yield potential during two years’ field experiments in the UK. These results confirm the notion that the *Ppd-1a* effect on yield and yield-related traits is pleiotropic and ambiguous, and under less stressful environmental conditions, early heading cultivars might suffer from a yield penalty in comparison to slowly developing ones [85]. Our results confirmed these observations since carriers of the *Ppd-1a*+*Vrn-A1a* genotype showed the lowest values of GNS and GWS among all studied NILs. Noteworthy, in the case of some NILs studied here, the negative effect of *Ppd-D1a* presence might be compensated by an increase in spike fertility via a higher number of grains per spikelet, which is considered to be an important factor contributing to grain yield improvement [66,79,92,93]. These results are in accordance with the results of [16]. Interestingly, the *Ppd-D1a* allele is frequently found in the southern region of Europe, Turkey, Pakistan, and Japan and goes in combination with either *Vrn-B1* or *Vrn-D1* dominant alleles. The *Ppd-D1b* allele is characteristic of cultivars thriving in the Central, Western, and Northern regions of Europe, Western Siberia, the Northern regions of Canada, and the USA, where the majority of cultivated wheats are carriers of *Vrn-A1* and *Vrn-A1*+*Vrn-B1* genotypes [14,40,44,55,57,87]. However, despite recognized pleiotropic effects on wheat yield, the frequency of *Ppd-1a* allele presence has generally increased worldwide, even in the regions historically known for breeding and planting photoperiod-sensitive cultivars [55,94,95].

A key concern for wheat breeders is to adjust the plant time of heading in such a way that it fits into a safe period when the risk of heat stress or freezing events is minimal, while also taking into account the rainfall pattern [10]. In our study, the effect of different *Vrn-1* and *Ppd-1* allelic combinations on adaptive and agronomic traits of common wheat was investigated in a near-ideal environment mimicking a prolonged growing season with favorable irrigation conditions and a long photoperiod. However, further validation of these NILs performance during field-based experiments in different agro-climatic zones and under different photoperiod conditions would be interesting. Information obtained in this study may be used by breeders to accelerate the breeding process towards adaptability of wheat to changing environmental conditions.

## 4. Materials and Methods

### 4.1. Plant Material and Growth Conditions

Five sets of NILs for the VRN-1 loci were investigated in this study. A description of the lines is provided in Table 1. Four NIL sets in the genetic background of winter wheat cultivars, viz., Skorospelka 3b (Sk3b), Mironovskaya 808 (M808), Priboi (Pr), and Johnes Fife (JF), were used in this study [61]. Spring wheat cultivars Triple Dirk D (TDD, Vrn-A1a), Triple Dirk B (TDB, Vrn-B1), and Triple Dirk E (TDE, Vrn-D1) were used by Stelmakh and Avsenin [61] as donors of Vrn-1 alleles during the development and characterization of the aforementioned NILs. TDD, TDB, and TDE were also used in our study since they are part of the NILs series carrying different alleles for Vrn-1 in the genetic background of the Triple Dirk (TD) cultivar. TDC (vrn-A1, vrn-B1, vrn-D1), a winter wheat cultivar, was used as a standard [59,60,61].

Before planting, the seeds of the studied NILs were germinated on filter paper in water on Petri dishes. Ten to twenty plants for every NIL were sown in the hydroponic greenhouse of the Institute of Cytology and Genetics, SB RAS, during the years 2022 and 2023. Plants were grown in a greenhouse at an air temperature of 23–25 °C under a 16 h daylight regimen. Provision of a nutrient solution consisting of water and fertilizer as well as phytosanitary protection measures were carried out according to local practice.

Four sets of NILs (near-isogenic lines) were obtained from the VIR collection (St. Petersburg, Russia): (I) Skorospelka 3b, (II) Mironovskaya 808, (III) Priboi, and (IV) Johnes Fife (K-60639-K-60666) and cv. Sonora 64 (K-47942).

NILs: TD D (AUS90069), TD B (AUS90066), TD E (AUS90322), and TD C (ICG collection) were kindly provided by Prof. A.T. Pugsley (Agricultural Research Institute, Wagga, NSW, Australia).

Chinese Spring were kindly provided by Prof. E.R. Sears from Missouri University (USA).

### 4.2. Plant Phenotyping

Heading time (HT) in days, tiller number (TN), and plant height (PH) in cm were determined in the greenhouse for each plant. Spike morphology traits were evaluated after harvest. Main spike length (SL) in cm, spikelet number per spike (SNS), number of grains per spike (GNS), spike fertility (SF, calculated in grain number per number of spikelet, GNS/SNS), and grain weight (GWS) per spike (in g) were determined.

### 4.3. Genomic DNA Extraction and PCR Analysis

Genomic DNA extraction was performed with the HiPure SF Plant DNA kit (Magen Biotechnology Co., Ltd., Guangzhou, China) according to manufacturer instructions. Briefly, fresh plant leaves were collected and immediately frozen in liquid nitrogen. 80 mg of frozen leaf material were used for extraction. DNA concentration and quality were checked with NanoDrop2000 (Thermo Scientific, Waltham, MA, USA). DNA samples with a 260 nm/280 nm ratio equal to or higher than 1.8 were considered suitable for further PCR analysis.

A study of Vrn-1 and Ppd-1 loci polymorphisms was performed with a set of previously described allele-specific primers (Appendix A). For *Ppd-D1* alleles detection (primer set: *Ppd-D1*-*F1*/*Ppd-D1-R1*/*Ppd-D1-R2* [51]), the following “touchdown” PCR protocol was used: initial denaturation at 96 °C for 5 min, then 10 cycles (96 °C for 30 s; 64 °C for 30 s with the temperature decreasing by 1 °C in each subsequent cycle; 72 °C for 90 s), 29 cycles (96 °C for 30 s; 54 °C for 30 s; 72 °C for 90 s), and final elongation at 72 °C for 5 min. In this protocol, all three primers were simultaneously used in a single PCR reaction. DNA samples of cultivars Sonora64 and Chinese Spring were used as controls (C2 and C1) for *Ppd-B1a* and *Ppd-B1c* detections, respectively. The same cultivars were used as positive and negative controls, respectively, during *Ppd-D1* allele verification. PCR reactions were performed in 20 μL with 1×HS Taq PCR Color master mix (Biolabmix, Novosibirsk, Russia), 0.5 μM of each primer, and 30 ng to 60 ng of genomic DNA. The reaction was run on a Bio Rad T100 Thermal Cycler (Bio-Rad, Hercules, CA, USA). The PCR products were separated in 1–3% agarose gels depending on the PCR product size and visualized under UV light after staining with ethidium bromide. DNA marker Step100long (Biolabmix, Novosibirsk, Russia) was used for PCR product size identification.

### 4.4. Data Analysis

Phenotypical data obtained from different years were checked for homogeneity with the PERMANOVA method (function adonis2 of the statistical package of vegan language R, statistical software R (Version 4.3.1)). The assessment showed that there was no statistically significant difference between the two years for the combined dependent variables (F = 0.8855, *p* = 0.4028). It was found that the influence of the year as a factor of variation in signs is quite small: R2 = 0.00269. Therefore, phenotypical data collected during the 2022 and 2023 years of cultivation of NILs were legitimately merged and analyzed as a single data set for all types of analyses.

Standardized data were used for the analysis. Data standardization and PCA analysis were performed as previously described by Smolenskaya et al. [15], with minor modifications. Briefly, quantitative phenotypical and agronomic traits (HT, PH, SL, etc.) as well as qualitative characteristics (presence of awns, allelic variations of *Vrn-1* and *Ppd-1*) were used for statistical processing. The data matrices of qualitative traits (genes alleles, presence of awns) were transformed into binary values in the following way: for the *Vrn-1* gene, 1 stood for the spring allele and 0 for the winter allele; for the *Ppd-1* gene, 1 stood for the photoperiod insensitive and 0 for the photoperiod sensitive allele; for awnedness, 1 stood for awnless isolines and 0 stood for awned ones. The complete data matrix (number of plants*number of traits = 330*14) was analyzed with PCA to identify the structure of variance and the associations between the studied variables. Principal component computation was performed with singular value decomposition (SVD procedure) by using the function prcomp in the R statistical package (statistical software R (Version 4.3.1)). Descriptive statistics and analysis of variance (ANOVA) of the collected data on various traits were performed, and the results were considered significant if *p* < 0.05. A Bonferroni post-hoc test at 5% probability was used to reveal groups where ANOVA indicated a significant difference. Differences were considered insignificant if the Bonferroni test was not passed.

## Figures and Tables

**Figure 1 plants-13-01453-f001:**
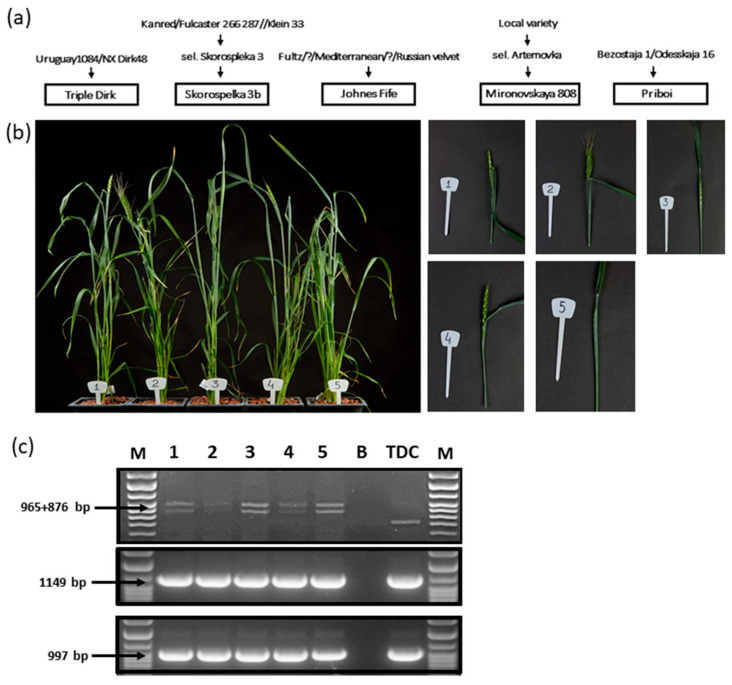
(**a**) Pedigree of the NILs recurrent parental cultivars [69]. (**b**) Plant phenotypic comparison between NILs with the *Vrn-A1* dominant allele of *Vrn-1*. Here and in the following panels, 1 stands for TDD, 2 for Sk3b (Vrn-A1), 3 for JF (Vrn-A1), 4 for M808 (Vrn-A1), and 5 for Pr (Vrn-A1). (**c**) Verification of the Vrn-1 alleles in the abovementioned lines. From left to right: DNA ladder, NILs, blank (B), and TDC (winter wheat cultivar) as a standard.

**Figure 2 plants-13-01453-f002:**
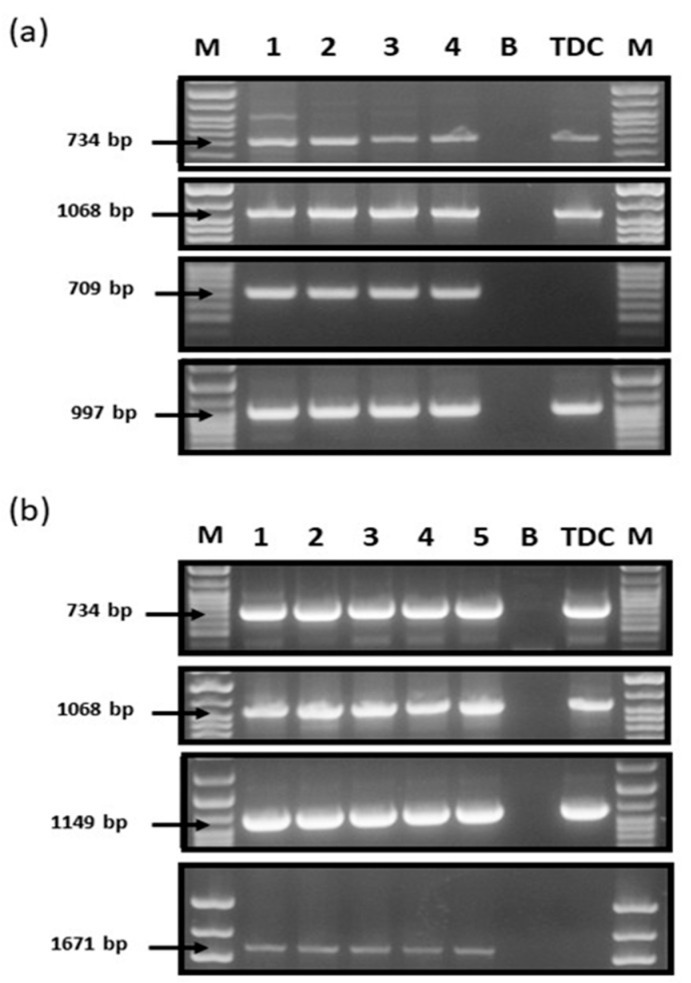
(**a**) Verification of *Vrn-1* alleles in NILs. Number 1 stands for TDB, 2 for Sk3b (Vrn-B1), 3 for M808 (Vrn-B1), 4 for Pr (Vrn-B1), (B) for blank, and TDC as standard. (**b**) Verification of *Vrn-1* alleles in NILs. 1 stands for TDE, 2 for Sk3b (Vrn-D1), 3 for JF (Vrn-D1), 4 for M808 (Vrn-D1), 5 for Pr (Vrn-D1), blank (B), and TDC as a standard.

**Figure 3 plants-13-01453-f003:**
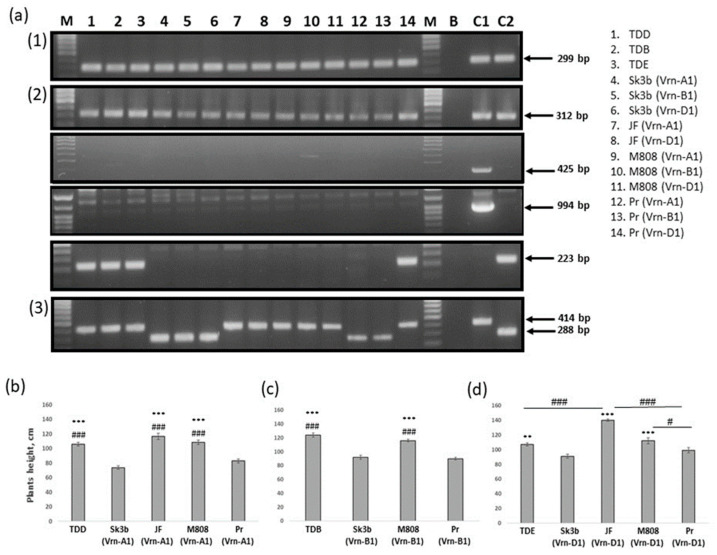
PCR amplification performed with allele specific primers to detect allele diversity of (**a1**) *Ppd-A1*; (**a2**) *Ppd-B1*; and (**a3**) *Ppd-D1* (M—DNA ladder Step100Long (Biolabmix, Russia); B—water; C1—Chinese Spring; C2—Sonora64). Comparison of the PH between different NILs with the following *Vrn-1* genotypes: (**b**) *Vrn-A1 vrn-B1 vrn-D1*; (**c**) *vrn-A1 Vrn-B1 vrn-D1*; and (**d**) *vrn-A1 vrn-B1 Vrn-D1*. In all graphs, values are presented as the mean ± SE. ANOVA with Bonferroni’s post-hoc test was performed (** *p* < 0.01, *** *p* < 0.001, * used to pinpoint significance for all comparisons between NILs (Sk3b) and the rest, ^#^
*p* < 0.05, for (**b**,**c**) ^###^
*p* < 0.001, ^#^ used to pinpoint significance for all comparisons between NILs (Pr) and/or TDE and the rest).

**Figure 4 plants-13-01453-f004:**
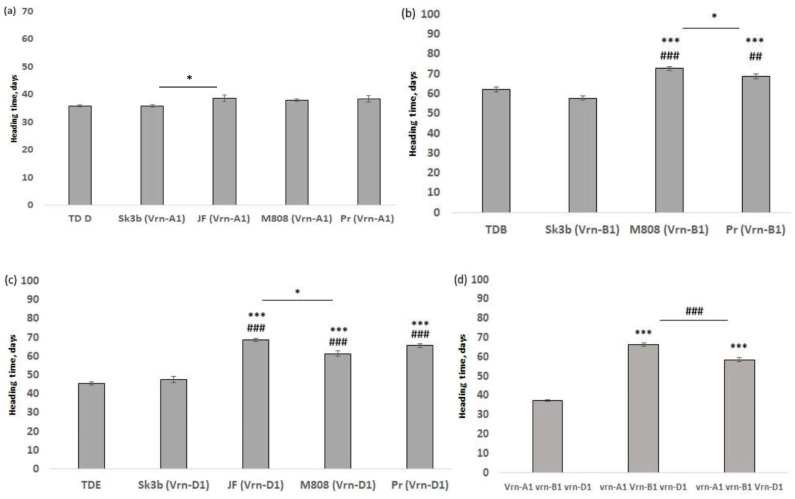
(**a**) Comparison of days to heading between the abovementioned NILs. Values are presented as the mean ± SE. ANOVA with Bonferroni’s post-hoc test was performed, * *p* < 0.05; (**b**) Comparison of days to heading between NILs; (**c**) Comparison of days to heading between NILs; (**d**) Comparison of days to heading between plants with the following genotypes: Vrn-A1, vrn-B1, vrn-D1, vrn-A1, Vrn-B1, vrn-D1, vrn-A1, vrn-B1, and Vrn-D1. In all graphs, values are presented as the mean ± SE. ANOVA with Bonferroni’s post-hoc test was performed. * *p* < 0.05, *** *p* < 0.001, * used to point significance for all comparisons between NILs (Sk3b) and the rest, ^##^
*p* < 0.01, ^###^
*p* < 0.001 when ^#^ used to point significance for all comparisons between NILs (TD) and the rest.

**Figure 5 plants-13-01453-f005:**
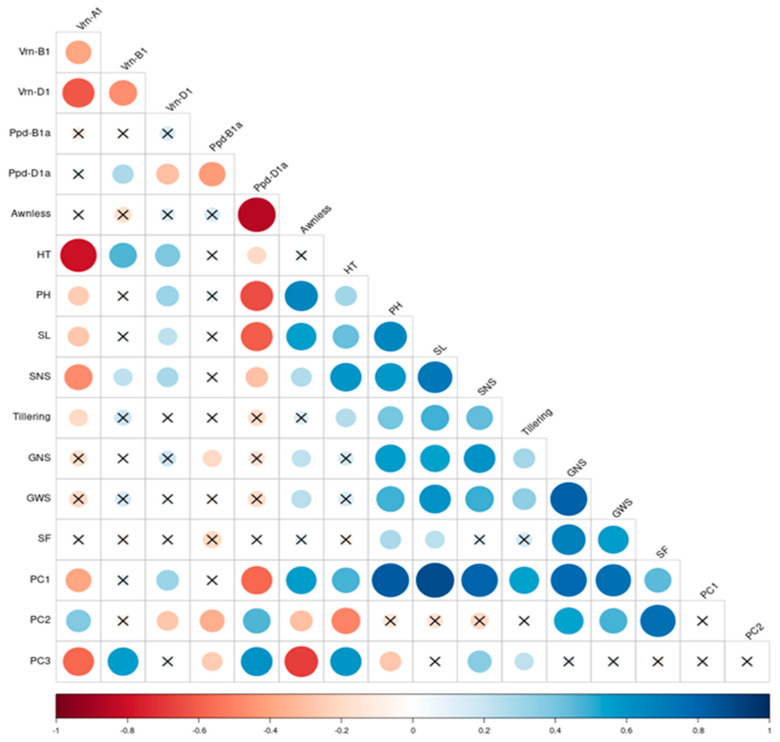
Pearson’s correlations of the principal components with determined quantitative and qualitative traits and the correlation of these traits with each other. Significance at *p* < 0.05 was determined with the Bonferroni test. All non-significant correlations are assigned with a cross (X).

**Table 1 plants-13-01453-t001:** *Vrn-1* allelic combination and breeding history of NILs used in this study according to [58,61,62].

NILs Original Name	Abbreviations Used in This Study	Presumed *Vrn-1* Allelic Composition	Presumed *Ppd-1* Allelic Composition	Breeding History (Donor//Recurrent cv)
Skorospelka 3b (*Vrn1*)	Sk3b (*Vrn-A1*)	*Vrn-A1 vrn-B1 vrn-D1*	*Ppd-A1b Ppd-B1b Ppd-D1a*	Triple Dirk D//9 * Skorospelka 3b
Skorospelka 3b (*Vrn2*)	Sk3b (*Vrn-B1*)	*vrn-A1 Vrn-B1 vrn-D1*	*Ppd-A1b Ppd-B1b Ppd-D1a*	Triple Dirk B//9 * Skorospelka 3b
Skorospelka 3b (*Vrn3*)	Sk3b (*Vrn-D1*)	*vrn-A1 vrn-B1 Vrn-D1*	*Ppd-A1b Ppd-B1b Ppd-D1a*	Triple Dirk E//9 * Skorospelka 3b
Johnes Fife (*Vrn1*)	JF (*Vrn-A1*)	*Vrn-A1 vrn-B1 vrn-D1*	*Ppd-A1b Ppd-B1b Ppd-D1b*	Triple Dirk D//9 * Johnes Fife
Johnes Fife (*Vrn3*)	JF (*Vrn-D1*)	*vrn-A1 vrn-B1 Vrn-D1*	*Ppd-A1b Ppd-B1b Ppd-D1b*	Triple Dirk E//9 * Johnes Fife
Mironovskaya 808 (*Vrn1*)	M808 (*Vrn-A1*)	*Vrn-A1 vrn-B1 vrn-D1*	*Ppd-A1b Ppd-B1b Ppd-D1b*	Triple Dirk D//9 * Mironovskaya 808
Mironovskaya 808 (*Vrn2*)	M808 (*Vrn-B1*)	*vrn-A1 Vrn-B1 vrn-D1*	*Ppd-A1b Ppd-B1b Ppd-D1b*	Triple Dirk B//9 * Mironovskaya 808
Mironovskaya 808 (*Vrn3*)	M808 (*Vrn-D1*)	*vrn-A1 vrn-B1 Vrn-D1*	*Ppd-A1b Ppd-B1b Ppd-D1b*	Triple Dirk E//9 * Mironovskaya 808
Priboi (*Vrn1*)	Pr (*Vrn-A1*)	*Vrn-A1 vrn-B1 vrn-D1*	*Ppd-A1b Ppd-B1b Ppd-D1a*	Triple Dirk D//9 * Priboi
Priboi (*Vrn2*)	Pr (*Vrn-B1*)	*vrn-A1 Vrn-B1 vrn-D1*	*Ppd-A1b Ppd-B1b Ppd-D1a*	Triple Dirk B//9 * Priboi
Priboi (*Vrn3*)	Pr (*Vrn-D1*)	*vrn-A1 vrn-B1 Vrn-D1*	*Ppd-A1b Ppd-B1a Ppd-D1b*	Triple Dirk E//9 * Priboi
Triple Dirk D	TDD	*Vrn-A1 vrn-B1 vrn-D1*	*Ppd-A1b Ppd-B1a Ppd-D1b*	Winter Minflor/3–4 * TD
Triple Dirk B	TDB	*vrn-A1 Vrn-B1 vrn-D1*	*Ppd-A1b Ppd-B1a Ppd-D1b*	Winter Minflor/3–4 * TD
Triple Dirk E	TDE	*vrn-A1 vrn-B1 Vrn-D1*	*Ppd-A1b Ppd-B1a Ppd-D1b*	Loro/3–4 * TD

* backcrosses.

**Table 2 plants-13-01453-t002:** Mean values of phenology- and yield-related traits recorded for NILs used in this study. HT, heading time; PH, plant height; SL, spike length; SNS, spikelet number per spike; GNS, grain number per spike; GWS, grain weight per spike; SF, spike fertility. Mean values and standard error (SE) for various traits were calculated; ns indicates non-significant.

NILs	HT, Days	PH, cm	SL, cm	SNS	Tillering	GNS	GWS, g	SF
TDD	35.9 ± 0.4	106 ± 2.4	8.1 ± 0.2	14.3 ± 0.3	6.8 ± 0.5	24.8± 1	1.16 ± 0.05	1.8 ± 0.03
TDB	61.9 ± 1.2	124 ± 2.8	9± 0.3	20 ± 0.8	10.2 ± 1.1	24.7 ± 4.2	1.1 ± 0.18	1.5 ± 0.2
TDE	45.6 ± 0.9	107.6 ± 2	8.6 ± 0.1	15.5± 0.4	6 ±0.4	23.5 ± 1.4	1.23 ± 0.1	1.5 ± 0.1
ANOVA	*p* < 0.001	*p* < 0.001	*p* < 0.05	*p* < 0.001	*p* < 0.001	ns	ns	*p* < 0.05
Sk3b (Vrn-A1)	35.8 ± 0.4	73.8 ± 2.4	6.2 ± 0.2	13 ± 0.3	4.3 ± 0.3	24.1 ± 1.3	0.79 ± 0.1	1.8 ± 0.1
Sk3b (Vrn-B1)	57.7 ± 0.9	91.8 ± 3	6.7 ± 0.2	14.1 ± 0.6	5.9 ± 0.6	27.4 ± 2.3	1.1 ± 0.1	1.9 ± 0.1
Sk3b (Vrn-D1)	47.6 ± 1.7	91.3 ± 2.7	6.8 ± 0.2	14.3 ± 0.4	6.3 ± 0.7	29.9 ± 1.6	1.19 ± 0.1	2.1 ± 0.1
ANOVA	*p* < 0.001	*p* < 0.001	*p* < 0.05	ns	*p* < 0.05	*p* < 0.05	*p* < 0.001	ns
JF (Vrn-A1)	38.7 ± 1.2	117 ± 4.3	8.4 ± 0.3	14.8 ± 0.5	5.9 ± 0.9	35.6 ± 1.9	1.27 ± 0.1	2.3 ± 0.1
JF (Vrn-D1)	68.5 ± 0.8	140 ± 1.5	10.5 ± 0.2	23.2 ± 0.5	9.3 ± 1	43.4 ± 2.3	1.35 ± 0.1	2 ± 0.1
ANOVA	*p* < 0.001	*p* < 0.001	*p* < 0.001	*p* < 0.001	*p* < 0.05	*p* < 0.05	ns	*p* < 0.01
M808 (Vrn-A1)	38 ± 0.5	108.6 ± 3	9.4 ± 0.2	14.5 ± 0.2	7.5 ± 0.6	24.2 ± 1.5	1.13 ± 0.1	1.7 ± 0.1
M808 (Vrn-B1)	72.7 ± 1.1	116.4 ± 2.3	11 ± 0.3	19.5 ± 0.5	10.5 ± 1.2	34.3 ± 2.4	1.58 ± 0.2	2 ± 0.1
M808 (Vrn-D1)	61.3 ± 1.5	112.4 ± 3.9	10.3 ±0.4	18.2 ± 0.7	8.7 ± 1.5	31.9 ± 3.6	1.39 ± 0.2	1.8 ± 0.5
ANOVA	*p* < 0.001	ns	*p* < 0.01	*p* < 0.001	ns	*p* < 0.01	ns	ns
Pr (Vrn-A1)	38.5 ± 1.1	83.4± 2.5	7.5 ± 0.2	15.6 ± 0.2	7.8 ± 0.6	21.9 ± 1	0.78 ± 0.1	1.5 ± 0.1
Pr (Vrn-B1)	68.5 ± 1.2	90.2 ± 1.9	8.5 ± 0.3	20.2 ± 0.8	9.2 ± 0.5	26.7 ± 2.1	1.25 ± 0.1	1.5 ± 0.1
Pr (Vrn-D1)	65.7 ± 1	99.7 ± 3.6	9.3 ± 0.3	17.8 ± 0.6	9.5 ± 0.9	23.3 ± 2.9	0.94 ± 0.1	1.6 ± 0.1
ANOVA	*p* < 0.001	*p* < 0.001	*p* < 0.001	*p* < 0.001	ns	ns	*p* < 0.05	ns

**Table 3 plants-13-01453-t003:** Structural changes of the genes that control the duration of the vegetation period of the studied accessions.

	PC1	PC2	PC3	PC4	PC5	PC6	PC7	PC8	PC9	PC10
*Vrn-A1a*	−0.112	0.166	−0.280	0.205	−0.253	0.052	−0.151	0.083	0.211	−0.032
*Vrn-B1*	0.022	−0.035	0.242	0.086	−0.258	−0.377	0.271	−0.290	0.089	−0.018
*Vrn-D1*	0.090	−0.131	0.038	−0.291	0.511	0.325	−0.120	0.207	−0.300	0.050
*Ppd-B1a*	0.004	−0.183	−0.140	0.080	0.261	0.310	0.714	−0.171	0.357	0.041
*Ppd-D1a*	−0.203	0.258	0.364	0.043	−0.103	0.054	−0.158	−0.162	−0.093	−0.093
Awnless	0.264	−0.236	−0.554	−0.068	−0.149	−0.330	−0.062	0.002	−0.129	0.472
HT	0.220	−0.379	0.479	−0.287	0.091	−0.416	0.051	0.061	0.016	0.050
PH	0.387	−0.108	−0.226	−0.040	0.147	0.009	−0.250	−0.648	−0.027	−0.523
SL	0.410	−0.121	−0.040	0.110	−0.239	0.047	0.055	0.584	0.075	−0.519
SNS	0.372	−0.149	0.282	−0.029	−0.283	0.395	−0.200	−0.053	0.448	0.267
Tillering	0.253	−0.038	0.177	0.840	0.352	−0.080	−0.116	0.018	−0.121	0.193
GNS	0.365	0.400	0.062	−0.169	−0.055	0.215	−0.091	−0.133	0.003	0.327
GWS	0.353	0.354	0.053	−0.008	−0.218	0.072	0.469	−0.019	−0.535	−0.015
SF	0.204	0.565	−0.058	−0.148	0.408	−0.391	0.020	0.159	0.443	−0.044
SD	2.14	1.33	1.23	0.90	0.74	0.67	0.65	0.56	0.495	0.39
Variability, %	41.58	16.06	13.84	7.34	5.01	4.05	3.88	2.91	2.23	1.415
Cumulative, %	41.58	57.65	71.49	78.83	83.84	87.89	91.77	94.68	96.91	98.33

## Data Availability

Data are contained within the article and Appendix A.

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
