# Peer review of "Analysis of the Effects of the Vrn-1 and Ppd-1 Alleles on Adaptive and Agronomic Traits in Common Wheat (Triticum aestivum L.)"

_plants, 2024, doi:10.3390/plants13111453_

Round 1

Reviewer 1 Report

Comments and Suggestions for Authors

This paper analyses the effects of the Vrn-1 and Ppd-1 alleles on the adaptive and agronomic traits in common wheat using five sets of near-isogenic lines (NILs). The paper is of interest and well-prepared. However, some points need to be considered. I have made my comments and edits directly in the MS file, the major points are:

 ·       In the abstract, mentioning the evaluation conditions (experimental condition) would be better. I also think that before mentioning the results of the PCA, it is necessary to briefly mention the effects of different alleles on different traits, in the same order as the results section, as this is the main topic of the study.

·       The introduction is relatively long, but I think it provides comprehensive information about the topics of the study.

·       In the results section, I think there is a mistake in Table 1, please check. You can use abbreviations (e.g. the standard deviation) and explain them in the footnote. As it is hard to track, I think Table 2 can be presented better. Please also pay attention to how the figures are written in the tables. Please improve the resolution of the figures.

·       I think some rearrangement of the results presentation is needed. It is hard to follow when you go back and forth in the results presentation.

·       The discussion section is good; however, minor editing might be required (please see the attached file). Please avoid extensively using impressive words, such as interestingly. Although it might be accepted by the journal, I think saying for example by Plotnikov et al. [1] could be better than by [1].

·        In the M&M, what was the reason for the relatively short duration to the heading time in most NILs, even though you mentioned that your experimental setup mimics the favorable conditions of the prolonged growing season?

·       Please consistently use either cultivar or variety, unless it is meant to differentiate between the two.

Additional comments and edits can be found in the MS file

Comments on the Quality of English Language

Minor English editing might be required.

Author Response

Please, see attachment

Reviewer 2 Report

Comments and Suggestions for Authors

Please give figures with higher resolution, specially for Figure 1b, Figure 5, Figure 6a and 6b

Author Response

Please, see attachment

Reviewer 3 Report

Comments and Suggestions for Authors

In this study, five sets of near-isogenic lines (NILs) were studied with an aim to investigate the effect of VRN-1 (response to vernalization) and PPD-1 (response to photoperiod) on wheat duration of vegetative period and yield-related traits. Molecular markers were employed to assess their allelic composition at VRN-1 and PPD-1 loci. To reduce data dimensionality and to inspect relationships between traits, principal component analysis (PCA) was used. However, the interpretation and description of many results in the manuscript are too unclear.

The major concerns are as following:

1.     The purpose of the study was to evaluate the effects of different combinations of Vrn-1 and Ppd-1 alleles on heading date and major agronomic traits of common wheat. However, PCA analysis results were largely mentioned in the abstract, but no specific alleles combination was specified to be beneficial to heading date and major agronomic traits of common wheat.

2.     The author mentioned the detailed information of 14 near-isogenic lines in Table1, but in fact the information of Days to heading and mean± standard deviation was missing. How to select from 14 near-isogenic lines to 5 near-isogenic lines is also not indicated in the manuscript.

3.     Traditional PCA analysis requires independent variables to be continuous variables. In Table 3, is it reasonable to integrate the qualitative traits (Vrn-1 gene, 1 stood for the spring allele and 0 for the winter allele) and quantitative traits (HT, PH, SL, etc.) PCA analysis together? If the author thinks it is reasonable, is there any published work for reference?

4.     The author reached this conclusion none of studied genes had strong correlations with yield-related traits through Pearson correlation analysis and PCA analysis. I think the evidence was insufficient.

5.     In my opinion, this work lacks proper logic from experimental design to experimental method selection and result presentation.

Author Response

Please, see attachment.

Round 2

Reviewer 3 Report

Comments and Suggestions for Authors

According to the revised version submitted by the author, all my previous questions have been answered.